# Solid-State Fermented Cereals: Increased Phenolics and Their Role in Attenuating Liver Diseases

**DOI:** 10.3390/nu17050900

**Published:** 2025-03-04

**Authors:** Ganesha Yanuar Arief Wijaya, Andrea Vornoli, Lucia Giambastiani, Maria Digiacomo, Marco Macchia, Bartłomiej Szymczak, Marta Wójcik, Luisa Pozzo, Vincenzo Longo

**Affiliations:** 1Doctoral School in Life Sciences, University of Siena, 53100 Siena, Italy; g.wijaya@student.unisi.it; 2Department of Pharmacy, University of Pisa, Via Bonanno, 56126 Pisa, Italy; maria.digiacomo@unipi.it (M.D.); marco.macchia@unipi.it (M.M.); 3CNR-IBBA, Institute of Agricultural Biology and Biotechnology, National Research Council, Via Moruzzi 1, 56121 Pisa, Italy; andrea.vornoli@cnr.it (A.V.); lucia.giambastiani@ibba.cnr.it (L.G.); vincenzo.longo@cnr.it (V.L.); 4Interdepartmental Research Center “Nutraceuticals and Food for Health”, University of Pisa, 56100 Pisa, Italy; 5Sub-Department of Pathophysiology, Department of Preclinical of Veterinary Sciences, Faculty of Veterinary Medicine, University of Life Sciences in Lublin, Akademicka 12, 20-033 Lublin, Poland; bartlomiej.szymczak@up.lublin.pl; 6Veterinary Oncology Lab., Department of Epizootiology and Clinic of Infectious Diseases, Faculty of Veterinary Medicine, University of Life Sciences in Lublin, Głęboka 30, 20-612 Lublin, Poland; marta.wojcik@up.lublin.pl

**Keywords:** fermented cereals, phenolic compounds, liver diseases, hepatoprotective effects, solid-state fermentation

## Abstract

Liver diseases, a leading cause of global mortality, necessitate effective dietary strategies. Fermented cereals, traditionally recognized for benefits in glucose regulation, lipid profiles, and antioxidant activity, hold potential for managing conditions such as type 2 diabetes, hypertension, and obesity. However, their specific impact on liver health requires further investigation. Fermentation, particularly solid-state fermentation (SSF), enhances the bioavailability of beneficial compounds, including phenolics. This review summarizes recent studies on the phenolic content of fermented cereals, highlighting variations based on microbial strains and cereal types. It examines the hepatoprotective effects of these phenolics, drawing on *in vivo* and *in vitro* research. Furthermore, the review explores recent findings on the impact of fermented cereals on liver health and related diseases. This work provides a foundation for future research exploring fermented cereals as a dietary intervention for liver disease prevention and management.

## 1. Introduction

In recent years, the prevalence of liver diseases has risen significantly. According to the World Health Organization (WHO), liver diseases, including cirrhosis and other chronic conditions, were responsible for approximately 2 million deaths globally in 2023 [1]. Among these, liver cancer accounted for the majority, with about 830,180 deaths recorded globally in 2020. In Europe, liver cancer remains a significant health concern, particularly in Italy, where it caused approximately 9000 deaths in 2020 [2]. These alarming statistics underscore the importance of exploring effective dietary and therapeutic interventions to combat liver diseases.

Cereals significantly contribute to the world’s food supply as sources of protein, dietary fiber, minerals, and vitamins, accounting for over 50% of global daily caloric intake, thus emphasizing their critical role in food security and nutrition worldwide [3,4]. Besides their nutritional value, cereals are rich in bioactive compounds such as flavonoids, isoflavones, phenolic compounds, and hypotensive peptides with angiotensin-converting enzyme (ACE) inhibitory activity [5]. These properties make cereals ideal for developing fermented functional foods [6].

Traditional fermented cereals from various regions have been extensively documented for their health benefits. In Africa, staples such as poto-poto (Congo), kenkey (Ghana), incwancwa (South Africa), and togwa (Tanzania) are notable examples, all derived from maize [7]. Similarly, in South and Central America, chicha de jora and pozol have been well-documented for their nutritional advantages [3]. These foods are known to promote improved digestion, enhance nutrient absorption, and support gut health. The recent recognition of fermented cereal-based products has spurred innovation in the nutraceutical industry, exemplified by products such as Lisosan G, made in Italy from organic whole grains such as *Triticum aestivum*, which has gained attention as a functional fermented food with numerous health benefits [8,9,10,11,12].

Fermentation, an ancient and energy-efficient food processing method, has gained attention not only for its nutritional value, taste, flavor, texture, and longer shelf life but also for its health benefits. The health benefits of fermented food were first observed by Russian scientist Élie Metchnikoff, who attributed the long lifespan of Bulgarian peasants to their high consumption of fermented products [13]. Fermentation enhances the bioactive components of food by harnessing the growth and metabolic activities of microorganisms [14]. A diverse range of microbial strains, such as lactic acid bacteria (LAB), *Acetobacter*, *Micrococcaceae*, yeast, fungi, or a mixture of these microorganisms, are typically used in fermentation and are responsible for the biotransformation of substrates into phenolic compounds [3,7]. Depending on the type of microorganism used, microbial interaction with cereal dough components can enhance the levels of bioactive molecules with nutraceutical functions [15]. Studies have reported that the fermentation process may increase the production of free phenolic compounds through microbial hydrolysis [16,17].

The antioxidant activity of fermented cereals increases due to the structural breakdown of plant cells, leading to the release of various antioxidant compounds [17,18]. At the cellular level in the human body, these antioxidants can act as free radical terminators, metal chelators, singlet oxygen quenchers, or hydrogen donors to radicals [19]. Due to these physiological properties, multiple studies have investigated the mechanisms by which fermented foods regulate health and their detailed effects on various diseases.

Recent studies have highlighted various health benefits associated with fermented cereals. For example, Alharbi and colleagues demonstrated that fermented oats inhibited streptozotocin-induced type 2 diabetes in rats, showcasing significant potential in regulating glucose levels and improving lipid profiles [20]. Similarly, Amato and colleagues showed that LAB exhibit antioxidant and anti-inflammatory effects, providing robust protective actions against the neural vascular defects characteristic of diabetic retinopathy [11]. Fermented foods have also been reported to be beneficial in the treatment of hypertension [16]. Other documented health benefits include obesity prevention, prevention of carcinogenesis, reduction of allergies, and mitigation of osteoporosis [14,21].

While the beneficial effects of fermented cereals on various health conditions are well-documented, there is a noticeable research gap specifically focusing on their impact on liver health. However, some existing studies suggest potential benefits fermented cereals might offer for liver health. Given the urgent need for novel dietary options for managing liver diseases, this article aims to provide a comprehensive review of the potential of cereal-based fermented foods in fighting liver diseases. Drawing on a body of studies published over the past decade (2015–2025), this work underscores the critical need for further research in this field.

## 2. The Importance of Solid-State Fermentation (SSF) to Enhance the Phenolics of Cereals

### 2.1. The Influence of Fermentation Types

Plant-based foods such as cereals are rich in bioactive compounds crucial for long-term liver health [22]. Fermentation, especially solid-state fermentation, enhances phenolic compounds beneficial to health [23,24]. Compared to liquid state fermentation (LSF), SSF is more effective for extracting bioactive compounds due to its low cost, efficiency, and environmental friendliness [25]. This process enhances phenolic content by growing microorganisms in a low-water environment [26].

Most phenolic compounds in cereals bind to cell wall components such as cellulose, hemicellulose, lignin, and pectin, with brown rice, corn, and wheat containing the highest percentages of bound phenolics at 88%, 85%, and 75%, respectively [27]. These phenolics are attached via ether or ester bonds [28]. SSF enhances the phenolic content of cereals by exploring bacterial enzymes that break these bonds, releasing soluble phenolics, as shown in Figure 1 [18,29].

Significant increases from bound phenolics to soluble/free phenols have been reported. For instance, Shumoy and colleagues noted that the total phenolic content (TPC) in tef increased from 14–17% to 17–32% after fermentation, with a significant rise in free phenolics [30]. Structural, surface, and chemical changes indicating this bond-breaking process have been observed using scanning electron microscopy (SEM), atomic force microscopy (AFM), and Fourier-transform infrared spectroscopy (FTIR) [18,31,32,33]. Free-form phenolic compounds are crucial for health as they are more digestible, as shown in Figure 1, easily absorbed in the small intestine, and efficiently distributed throughout the body [34].

The enzymes that actively break down the structure of bound phenolics depend primarily on the type of microorganism used [35]. During SSF, microorganisms such as single-strain bacteria, fungi, or mixed strains of bacteria and fungi are used (Table 1). These microorganisms produce enzymes that enhance different phenolic compounds. Although the activity of microbial enzymes is not always observed and reported, some research highlights this enzymatic activity. For instance, Bei and colleagues reported high activities of α-amylase and xylanase during the fermentation of oats [31]. Purewal and colleagues recently demonstrated that enzymatic activities, particularly β-glucosidase, α-amylase, and xylanase, increased steadily during the fermentation of barley, peaking on day 10 and thereby enhancing TPC [36]. Similarly, enzymes such as amylase, xylanase, and β-glucosidase have been recorded to enhance the TPC of fermented oats in several studies [18,26].

### 2.2. The Influence of Microorganisms Used in Cereal Fermentation

The microorganisms used in cereal fermentation can have different effects on the release of phenolic compounds. For example, Bei and colleagues found that enzymes from *Monascus anka* increased the release of phenolic compounds such as ferulic acid, vanillic acid, chlorogenic acid, and sinapic acid but degraded other phenolics such as caffeic acid, *p*-coumaric acid, and the flavonoid rutin during fermentation [31]. To better understand this, various SSF processes in different cereals, utilizing different microorganisms and the specific phenolics they enhance, are compiled in Table 1, based on the literature from the past 10 years. Additionally, some studies have observed increases in phenolic content without identifying specific phenolics, such as those by Sandhu et al., Kang et al., Chu et al., Balli et al., and Figueiredo et al., which are not documented in this table [32,37,38,39,40]. The enhancement of the phenolic compounds was observed by different instruments, but mostly with HPLC, HPLC-DAD, UHPLC, or UPLC-Q-TOF-MS, which are known as fast detectors of components [41,42,43,44,45,46].

As seen in Table 1, LAB, especially *Lactobacillus*, are the most widely used microorganisms for the SSF of cereals. The *Lactobacillus* group has been reported to enhance important phenolic compounds during SSF of various cereals, such as *trans*-ferulic acid (*t*FA), *p*-coumaric acid (*p*CA), vanillic acid (VA), caffeic acid (CA), syringic acid (SYRA), and gallic acid (GA). Additionally, other common phenols were also observed in some studies such as chlorogenic acid (CGA), sinapic acid (SA), (−)-epicatechin (EP), quercetin (QU), rutin (RU), vitexin (VX), 4-hydroxybenzoic acid (4HBA), dihydroferulic acid (DHFA), 5-5′ diferulic acid (5-5DFA), 8-O-4′ diferulic acid (8O4DFA), 8-5′ benzofuran diferulic acid (BFDFA), γ-oryzanol (γ-O), benzoic acid (BA), 2,4-dihydroxybenzoic acid (2,4DHBA), 2,6-dihydroxybenzoic acid (2,6DHBA), and 3,4-dihydroxycinnamic acid (3,4DHCA) [41,42,43,44,47,48,49,50,51].

Each of these phenols may have potential benefits for liver health, which will be discussed in the next part of the review. Interestingly, among *Lactobacillus* group, we found one study by Fan and colleagues who reported the effects of *Lactiplantibacillus plantarum* dy-1 (formerly known as *Lactobacillus plantarum*) in regulating the lipid metabolism in an ex vivo study or *in vitro* model by using the gut–liver axis [52]. This result shows the potential of *Lactiplantibacillus* fermentation for liver health as the gut–liver axis refers to the bidirectional relationship between the gastrointestinal tract and the liver, primarily mediated through the portal vein, which carries blood from the intestines to the liver. This connection plays a significant role in maintaining liver health and can influence the development and progression of liver diseases.

Among the enhanced phenols, ferulic acid and *p*-coumaric acid are the two most found in fermented cereals. Despite being naturally present in high concentrations in the cell walls of cereals (primarily found in the form of esters linked to polysaccharides such as arabinoxylans), the action of ferulic acid esterases, also known as hydroxycinnamoyl esterase, breaks down the complex cell wall structure to release ferulic acid and *p*-coumaric acid [53,54,55]. This makes ferulic acid and *p*-coumaric acid the two most enhanced phenolics found in fermented cereals. Besides enhancing the phenolics of fermented cereals, *Lactobacillus* is widely known to reduce polysaccharides (complex carbohydrates, including starches), thereby improving the health profile of fermented cereals by lowering sugar content. For instance, Sawangwan and colleagues demonstrated this in the SSF of rice bran [50]. Similarly, Zhu et al. reported a decrease in polysaccharide content during the fermentation of red rice, and Li et al. found that *Lactobacillus* aids in easier gelatinization of brown rice, making it more digestible and healthier. This lower sugar content also positively affects the liver health by reducing blood sugar levels and preventing liver fat accumulation [41,56].

In addition to *Lactobacillus*, which is the primary bacterium used in cereal fermentation, the fungi *Aspergillus* is also commonly employed. *Aspergillus* significantly enhances phenolic compounds in fermented cereals, particularly ferulic acid and *p*-coumaric acid [34,57,58]. *Aspergillus* species produce xylanase and ferulic acid esterase during SSF which are highly effective in breaking down hemicellulose and releasing phenolics [59,60,61]. Interestingly, *Aspergillus* also produces health-beneficial polysaccharides, such as β-glucans and galactomannans. These types of polysaccharides offer various health benefits, including hepatoprotective effects. For instance, β-glucans have been reported to prevent steatotic liver disease progression and liver ischemia, inhibit the growth of hepatocellular carcinoma (HepG2) liver cancer cells, improve liver regeneration after partial hepatectomy, and exert antitumor activity in liver cancer [49,62,63,64,65]. Galactomannans have been shown to prevent liver dysfunction, mitigate liver damage, and treat ethanol-induced fatty liver [66,67,68]. They also enhance immune response and increase protective efficacy, which positively affects liver health by protecting against infections and reducing inflammation [69].

Another commonly used group of fungi in cereal fermentation is *Rhizopus*, particularly *Rhizopus oryzae* and *Rhizopus oligosporus*. Unlike *Lactobacillus* and *Aspergillus*, the primary enzymes produced by *Rhizopus* are not ferulic acid esterase and xylanase. Instead, *Rhizopus* produces enzymes such as amylase, protease, lipase, phytase, cellulase, and hemicellulose [70,71,72]. As seen in Table 1, SSF with *Rhizopus* significantly enhances several important phenolics such as ferulic acid, vanillic acid, and caffeic acid. Interestingly, some studies using *Rhizopus* observed a decrease in or degradation of *p*-coumaric acid. Although the cause was not mentioned, it may be related to the activity of the enzymes involved, as *Rhizopus* may lack sufficient ferulic acid esterase and xylanase to efficiently release bound *p*-coumaric acid [45,51]. While direct evidence linking *Rhizopus* to liver health is limited, a study by Kameda et al. (2018) on tempeh found that tempeh prepared with *Rhizopus* had beneficial effects on liver function in rats fed high-fat diets [73]. Additionally, recent research reported that polysaccharides from *Rhizopus* can inhibit the malignant process of hepatocellular carcinoma *in vitro* and *in vivo*, indicating potential use for liver cancer therapy [74]. These studies highlight the potential of SSF with *Rhizopus* for liver health, and future research should focus on the effects of SSF with *Rhizopus* in cereals on liver health to provide more evidence.

Despite its advantages, SSF faces challenges in scaling up for industrial applications due to difficulties in maintaining uniform growth conditions, controlling mass and heat transfer, and ensuring culture homogeneity [75,76]. Additionally, enzyme production efficiency varies with different microbial strains and substrates, necessitating further optimization. Future research should focus on advanced biotechnological tools to engineer more efficient microbial strains, potentially leading to higher phenolic compound production. Combining SSF with other processing techniques, such as extrusion or high-pressure processing, could also further enhance the bioavailability of phenolic compounds [77,78,79]. Additionally, integrating genetic engineering to produce microorganisms with enhanced enzymatic capabilities is a promising research avenue [80,81]. Overall, optimizing SSF processes can significantly enhance cereal-based products by increasing their phenolic content, leading to the development of functional foods aimed at preventing various diseases.

**Table 1 nutrients-17-00900-t001:** SSF of fermented cereals with different microorganisms, antioxidant activity, and enhanced bioactive compounds.

Cereals	Type of Microorganism	Microorganisms	Enhanced Antioxidant Activity	Enhanced Phenolics	References
Rice bran	Fungi	*Rhizopus oryzae*	DPPH	tFA, CGA, VA, CA, GA, and 4HBA	[51]
Rice bran	Fungi	*Rhizopus oligosporus* and *Monascus purpureus*	DPPH and FRAP	TPC, tFA, SA, VA, CA, SYRA, and 4HBA	[45]
Oats	Fungi	*Cordyceps militaris*	DPPH, ABTS, and FRAP	TPC, tFA, *p*CA, LU, AP, GA, and avenanthramides	[44]
Oats	Fungi	*Monascus anka*	DPPH and ABTS	TPC, tFA, CA, and CGA	[46]
Whole wheat	Bacteria	*Lactobacillus plantarum* and *Lactobacillus hammesii*	-	TPC, tFA, VA, and DHFA	[47]
Maize	Fungi	*Aspergillus oryzae*, *Pleurotus ostreatus*, *Hericium erinaceus*	ORAC	tFA, *p*CA, 5-5DFA, 8O4DFA, and BFDFA	[34]
Rice bran	Bacteria	*Lactobacillus plantarum* MJM60383, *Lactococcus lactis* subsp. *lactis* MJM60392, *Lactobacillus fermentum* MJM60393, and *Lactobacillus paracasei* MJM60396	Fe^2+^ chelating activity, DPPH, and ABTS	tFA, *p*CA, and γ-O	[48]
Wheat bran	Fungi	*Aspergillus niger*	ORAC and cellular antioxidant activity (CAA)	tFA	[57]
Wheat bran	Bacteria	*Lactobacillus rhamnosus*	DPPH, ABTS, and FRAP	TPC, CA, 4HBA, *p*CA, tFA, and SA	[23]
Barley	Bacteria + Fungi	*Lactobacillus plantarum* and *Rhizopus oryzae*	DPPH, hydroxyl radical, ABTS, and FRAP	tFA, VA, and *p*CA	[49]
Rice bran	Bacteria	*Lactobacillus casei* and *Lactobacillus plantarum*	DPPH	TPC, tFA, and *p*CA	[50]
Rice bran	Fungi	*Aspergillus brasiliensis, Aspergillus awamori* and *Aspergillus sojae*	DPPH	TPC, TFC, tFA, *p*CA, SA, CA, and BA	[58]
Corn bran	Bacteria	*Lactobacillus reuteri* and *Lactobacillus plantarum*	DPPH, ABTS, and ORAC	TPC, tFA, and *p*CA	[18]
Red rice	Bacteria	*Lactiplantibacillus plantarum* dy-1	DPPH and ABTS	TPC, CA, SYRA, EP, and 2,4DHBA	[41]
Quinoa and black barley	Bacteria	*Lactobacillus kisonensis*	DPPH, ABTS, FRAP, and hydroxyl radical-scavenging activity/assay	TPC, TFC, tFA, *p*CA, 2,6DHBA, and 3,4DHCA	[42]
Pearl millet	Bacteria	*Lactobacillus sanfranciscensis* and *Lactobacillus pentosus*	DPPH, ABTS, FRAP, and ORAC	TPC, TFC, VA, GA, RU, QU, VX, and 4HBA	[43]

DPPH assay, assay with 2,2-diphenyl-1-picrylhydrazyl radical; FRAP, ferric-reducing antioxidant power; ABTS, 2,2′-azino-bis(3-ethylbenzothiazoline-6-sulfonic acid); ORAC, oxygen radical absorbance capacity; TPC, total phenolic content; TFC, total flavonoids content; *t*FA, *trans*-ferulic acid; CGA, chlorogenic acid; VA, vanillic acid; CA, caffeic acid; GA, gallic acid; 4HBA, 4-hydroxybenzoic acid; SA, sinapic acid; SYRA, syringic acid; *p*CA, *p*-coumaric acid; DHFA, dihydroferulic acid; 5-5DFA, 5-5′ diferulic acid; 8O4DFA, 8-O-4′ diferulic acid; BFDFA, 8-5′ benzofuran diferulic acid; γ-O, γ-oryzanol; BA, benzoic acid; EP, (−)-epicatechin; 2,4DHBA, 2,4-dihydroxybenzoic acid; 2,6DHBA, 2,6-dihydroxybenzoic acid; 3,4DHCA, 3,4-dihydroxycinnamic acid; QU quercetin; RU, rutin; VX, vitexin.

## 3. The Enhanced Antioxidant Activity and Phenolic Content of SSF Cereals for Liver Health

### 3.1. Effect of Enhanced Antioxidant Activity to Liver Health

As discussed in the previous section, the SSF process plays an important role in the bioactivity of fermented cereals by enhancing the antioxidant activity and phenolic compounds. The biological activities of fermented cereals, explored through extensive *in vitro* studies, underscore their potential to enhance liver health primarily through their robust antioxidant properties. Oxidative stress, a significant contributor to liver diseases such as viral hepatitis B or C, alcoholic fatty liver disease, metabolic dysfunction-associated steatotic liver disease (MASLD), metabolic dysfunction-associated steatohepatitis (MASH), liver fibrosis, cirrhosis, and hepatocellular carcinoma (HCC), arises from anaerobic metabolism and various pathological conditions [82,83]. The imbalance between antioxidants and oxidants in liver cells necessitates effective defense mechanisms, where functional foods such as fermented cereals play a pivotal role [84,85,86]. The accumulation of reactive oxygen species (ROS) and oxidative stress in liver cells can be triggered by various conditions, including drug-induced toxicity, excessive iron accumulation, dust exposure, pathological conditions such as ischemia-reperfusion injury, and exposure to ionizing radiation [87,88,89,90,91,92,93,94]. These factors, in turn, can exacerbate liver damage and impair its function.

Although the increase of TPC is a positive sign, TPC did not always include all antioxidants, and the increase of TPC did not necessarily indicate increased antioxidants [45]. This is because the Folin–Ciocalteu method that is commonly used to measure TPC is nonspecific and has limitations as some non-phenolic compounds can also react to the Folin–Ciocalteu reagent [44]. Therefore, *in vitro* studies have also highlighted the detection of enhanced components using several detectors such as high-performance liquid chromatography (HPLC) and ultra-high-performance liquid chromatography (UHPLC), as well as the enhancement of antioxidant capabilities of fermented cereals, as measured by various assays including 2,2-diphenyl-1-picrylhydrazyl (DPPH) radical-scavenging activity, ferric-reducing antioxidant power (FRAP), 2,2′-azino-bis (3-ethylbenzothiazoline-6-sulfonic acid) (ABTS) scavenging activity, and oxygen radical absorbance capacity (ORAC) (Table 1). Among the indicators or assays observed in past studies, DPPH activity is the most reported, being observed in almost all past studies (Table 1) because of its simplicity, convenience, and rapid detection [95]. Another simple assay used is FRAP, followed by the slightly more complex ABTS and ORAC assays. In general, past studies revealed an enhancement of the activity of these assays, indicating an increase in the antioxidant activity of various fermented cereals. While enhanced DPPH, FRAP, ABTS scavenging activity, and ORAC may not directly target liver health, the increased antioxidant activity they represent can significantly contribute to liver health. This is achieved by mitigating oxidative stress and preventing liver damage. Additionally, the enhanced antioxidant activity is associated with increased anti-inflammatory, antifibrotic, anticarcinogenic, detoxification, and lipid metabolism properties, all of which are beneficial for maintaining overall liver function and health [96].

As important as those assays related to the antioxidant activity, fermented cereals have also demonstrated significant results in other important assays such as angiotensin- I-converting enzyme (ACE) inhibition activity, which is beneficial for regulating blood pressure and potentially reducing liver fibrosis [97,98]. Another important assay is advanced glycation end-product (AGE) inhibition. AGE inhibition often overlaps with antioxidant activity, as some antioxidants can inhibit AGE formation [99]. This is particularly relevant for liver health, as AGE accumulation promotes the activation of hepatic stellate cells, which are responsible for fibrosis and can lead to cirrhosis [100].

As discussed in the previous section, the activity of various microbial enzymes activated during the SSF process can enhance specific components of cereals by releasing them from their bonds with the cell wall. This enzymatic action increases the availability of bioactive compounds, such as phenolic compounds, which contribute to the antioxidant properties and overall health benefits of fermented cereals. The effects of the phenolic compounds improved during the fermentation of cereals in past studies are summarized in Table 2.

### 3.2. Hepatoprotective Effects of Enhanced Phenolics

SSF has been shown to increase the phenolic content of cereals. Several of these phenolics have been extensively studied, primarily in animal models (mostly rats), and have demonstrated potential in promoting liver health by attenuating various liver diseases. Below, we summarize key phenolics identified in the literature and their potential liver health benefits, based on *in vitro* and *in vivo* studies.

#### 3.2.1. Ferulic Acid (FA)

Among the phenolic compounds, FA is the most commonly enhanced during cereal fermentation (Table 1). FA is known as the largest concentrated phenol in cereal, but it is mostly bound to the cell wall by ether bonds and can only be hydrolyzed in alkaline conditions and at high temperatures [101]. During fermentation, ferulic acid esterases are the most important enzymes of the gut microbiome, they release FA from feruloylated sugar ester conjugates, which occur naturally in grains, fruits, and vegetables [86]. FA has demonstrated significant antioxidant and anti-inflammatory effects [102,103,104]. Yin et al. found that FA inhibits malondialdehyde formation, reducing oxidative stress, and suppresses inflammation by decreasing the inflammatory cytokines such as tumor necrosis factor-α (TNF-α), interleukin-6 (IL-6), and nitric oxide (NO) levels [57].

As reported in Table 2, FA has been studied *in vivo* for its therapeutic effects on liver conditions in animal models, showing promise in attenuating liver fibrosis through various mechanisms [105,106]. Liver fibrosis, a chronic condition with high global mortality, results from liver injury and excessive extracellular matrix accumulation. A central factor in the development of liver fibrosis is the activation of hepatic stellate cells (HSCs), which contribute to fibrogenesis by producing extracellular matrix components (ECM) [107]. FA disrupts HSC activation through several mechanisms. It modulates the transforming growth factor β1 (TGF-β1) pathway, which activates HSCs and inhibits protein tyrosine phosphatase 1B (PTP1B) while promoting adenosine monophosphate-activated protein kinase (AMPK) phosphorylation [105,106]. Since AMPK inhibits HSC activation and PTP1B promotes it, FA’s ability to enhance AMPK activity and inhibit PTP1B suggests it effectively suppresses liver fibrosis progression through these combined mechanisms.

Besides liver fibrosis, recent studies have reported that FA can also alleviate various types of drug-induced liver injury, including methotrexate (MTX)-mediated liver injury and acetaminophen-induced liver injury. Drug-induced liver injury has long been a challenge in clinical treatment, as it can complicate drug therapy. Roghani et al. found that FA, at a dose of 100 mg/kg, effectively reduced MTX-mediated oxidative damage during cancer therapy by enhancing the endogenous antioxidant system. Additionally, FA was shown to mitigate acetaminophen-induced liver injury by alleviating oxidative stress and activating AMPK-mediated protective autophagy [87,88].

FA has also been reported to protect HepG2 cells and mouse liver from iron-induced liver injury [108]. Excess iron accumulation in the liver increases ROS production, leading to liver damage. In this study, FA protected against iron-induced liver injury and cellular dysfunction by activating the nuclear factor erythroid-2-related factor 2 (Nrf2) antioxidant pathway. *In vitro* studies with HepG2 cells also confirmed that FA increased levels of superoxide dismutase (SOD) and glutathione (GSH), contributing to ROS protection and iron-induced liver injury mitigation. Additionally, recent research proved that FA can alleviate gamma radiation-induced liver injury, commonly encountered in cancer treatment [94]. FA inhibited the janus kinase (JAK)/signal transducers and activators of the transcription (STAT) signaling pathway by upregulating Nrf2 to increase antioxidant synthesis, offering potential protection against radiation-induced liver damage during cancer therapy.

#### 3.2.2. *p*-Coumaric Acid (pCA)

The second most common phenolic compound enhanced as a product of the fermentation of cereals is *p*CA, as found in several studies in Table 1. Coumaric acid is a hydroxycinnamic acid compound, which has three isomers: m-coumaric acid, o-coumaric acid, and *p*-coumaric acid. *p*CA is the most abundant isomer of hydroxycinnamic acid in nature; it is a hydroxyl derivative that has been known for its strong antioxidant activity and protection against ROS, and also its immunomodulatory, anti-inflammatory, and antiangiogenic effects [109,110,111].

Like ferulic acid, *p*CA has also been reported to have potential for ameliorating various types of induced liver damage (Table 2). A study by Bal et al. demonstrated that *p*CA effectively mitigates fipronil (FPN)-induced liver injury. FPN, a newer insecticide, poses health risks due to its long half-life and bioaccumulation, leading to liver toxicity through increased lipid peroxidation, NO, and inflammatory cytokines when consumed continuously. In their study, *p*CA, at a dose of 100 mg/kg, effectively attenuated oxidative stress in mice by reducing serum liver enzymes, inflammatory cytokines, and myeloperoxidase activity, while increasing antioxidant enzyme levels [89].

Additionally, *p*CA has been reported to ameliorate radiation-induced liver damage (RILD), a significant issue often arising from radiotherapy in cancer treatment. RILD typically develops after exposure to radiation doses exceeding 30 Gy, with 5–10% of the radiation dose affecting the liver, a highly radiosensitive organ. In a study by Li et al., *p*CA supplementation at 100 mg/kg body weight effectively improved hematopoietic function in mice by inhibiting IR-induced hepatic sinusoidal congestion and steatosis, reducing Bcl-2-associated X (BAX) protein expression, and reversing changes in serum alanine aminotransferase (ALT) and aspartate aminotransferase (AST) activities [112].

Furthermore, *p*CA has been reported to protect against drug-induced liver injury, such as acetaminophen (AAP)-induced liver injury, as demonstrated in the study by Cha and colleagues [109]. AAP, a widely used analgesic and antipyretic, can cause oxidative stress in the liver, particularly when taken in excess or combined with alcohol. In their study, *p*CA was shown to inhibit ROS-dependent hepatic apoptosis and inflammation, thereby protecting against AAP-induced liver damage.

Lastly, recent studies have revealed that *p*CA may aid in treating liver damage induced by dust and ischemia-reperfusion (IR) injury [92]. Air pollution is a growing public health concern, contributing to various diseases, including liver conditions such as hepatic IR injury, which leads to oxidative stress and inflammation. In this study, pretreatment with *p*CA, significantly improved key variables, particularly by reducing the expression of HOX transcript antisense intergenic RNA (HOTAIR), offering partial protection against dust and IR-induced liver damage.

#### 3.2.3. Gallic Acid (GA)

Another common phenolic compound produced during cereal fermentation is GA (3,4,5-trihydroxybenzoic acid). Our literature review found numerous studies on GA in recent years. GA is recognized for its medicinal properties, particularly its antioxidant, anti-inflammatory, and hepatoprotective properties [113,114].

GA’s hepatoprotective effects have been extensively studied in recent years (Table 2). A study conducted by Rasool and colleagues highlighted its hepatoprotective and antioxidant activity in treating paracetamol-induced liver damage in mice. Paracetamol (acetaminophen) is a widely used antipyretic, but overdose can deplete cellular glutathione in the liver, as *N*-acetyl-p-benzoquinone imine (NAPQI) rapidly reacts with glutathione, leading to oxidative stress and hepatic cell damage. In Rasool et al.’s study, GA treatment, at a dose of 100 mg/kg, significantly restored antioxidant status and alleviated the progression of the inflammatory mediator TNF-α [115,116].

The potential of GA to treat drug-induced liver injury has also been demonstrated in ketamine-induced oxidative damage, diclofenac-induced liver toxicity, and isoniazid and rifampicin-induced liver injury. Ketamine, an anesthetic used in humans and animals, has pro-oxidant activity, inducing oxidative stress by increasing lipid peroxidation and depleting GSH levels. In Schimites et al.’s study, GA pre-treatment effectively prevented the increase in protein carbonyl (PC), a biomarker of oxidative stress, and preserved non-protein sulfhydryl (NPSH) groups, which were otherwise depleted by ketamine-induced oxidative damage [117]. Similarly, in Esmaeilzadeh and colleagues’ study, GA significantly elevated GSH, GPx, SOD, and catalase (CAT) levels, while reducing PC, AST, alkaline phosphatase (ALP), ALT, total bilirubin, malondialdehyde (MDA), serum interleukin-1 beta (IL-1β), and the gene expression of IL-1β, effectively treating diclofenac-induced liver toxicity [118]. The potential of GA in treating drug-induced liver injury was also demonstrated in a study conducted by Sanjay and colleagues on hepatotoxicity induced by the anti-tuberculosis and anti-HIV drugs isoniazid and rifampicin. GA effectively alleviated hepatocellular necrosis caused by oxidative stress and inflammation by upregulating the expression of endogenous antioxidants via the Nrf2 pathway and inhibiting nuclear factor kappa B (NF-κB)-mediated pro-inflammatory signaling [119].

Besides treating drug-induced liver injury, the potential of GA in treating liver fibrosis has also been observed in recent studies. Wang and colleagues demonstrated GA alleviates carbon tetrachloride (CCl_4_)-induced liver fibrosis in mice. CCl_4_ administration triggers hepatocyte death and inflammation, leading to HSC activation. As previously mentioned, HSC activation and excessive ECM deposition are key factors in liver fibrosis progression. In this study, GA effectively suppressed HSC activation by inhibiting matrix metalloproteinase-2 (MMP-2) and tissue inhibitor of metalloproteinase-1 (TIMP-1), both of which contribute to ECM remodeling and fibrosis [120]. GA’s effectiveness in alleviating liver fibrosis was also demonstrated in a study that investigated dimethylnitrosamine (DMN)-induced liver fibrosis, a common model using DMN as a toxin to study fibrosis. In this study, GA attenuated liver fibrosis by inhibiting the TGF-β/Small mother against decapentaplegic (Smad) signaling pathway, specifically by blocking Smad phosphorylation in liver cells [121].

Additionally, GA has been reported to attenuate various other liver conditions, including hepatic IR injury, lead (Pb)-induced liver toxicity, MASLD, type 2 diabetes-induced liver injury, and sepsis-induced liver injury [91,93,122,123,124,125].

#### 3.2.4. Caffeic Acid (CA)

CA, or 3,4-dihydroxycinnamic acid, is known for its strong antioxidant properties, and its hepatoprotective potential has been widely explored (Table 2) [126]. In Mu et al.’s study, CA was shown to alleviate liver ischemia-reperfusion injury, a condition often occurring after liver surgeries such as transplantation, by preventing the downregulation of sirtuin 3 (Sirt3) and enhancing mitochondrial respiratory chain (MRC) activity [127]. Further work by Mu et al. revealed that CA also protects post-transplant livers from oxidative damage through the inhibition of protein disulfide isomerase A3 (PDIA3)-dependent activation of nicotinamide adenine dinucleotide phosphate (NADPH) oxidase [128]. More recently, Mu et al. demonstrated that CA may help prevent obesity-associated MASLD by restoring gut microbiota balance and reducing lipopolysaccharide-mediated inflammation, thereby inhibiting dysregulation of lipid metabolism-related gene expression [129].

#### 3.2.5. Vanillic Acid (VA)

VA, or 3-methoxy-4-hydroxybenzoic acid, has recently gained attention for its hepatoprotective properties (Table 2) [130]. In a study by Punvittayagul et al., VA was found to offer protection against diethylnitrosamine (DEN) and 1,2-dimethylhydrazine (DMH)-induced hepatocarcinogenesis [131]. This protection was attributed to the upregulation of glutathione S-transferase alpha 5 (GSTA-5) and Nrf2 gene expression, along with a reduction in the number and area of hepatic GST-P-positive foci. VA also enhanced apoptosis activity by increasing caspase-3 and Bcl-2-associated death promoter (BAD) protein levels, contributing to its protective effects against liver cancer development [131].

Furthermore, VA has been reported to alleviate CCl₄-induced liver fibrosis, as demonstrated in the study by Qin et al. [130]. The study demonstrated that VA effectively inhibits the activation, proliferation, and migration of HSCs, which are key players in the progression of liver fibrosis. This protective effect is mediated by VA’s ability to inhibit autophagy in HSCs through the macrophage migration inhibitory factor (MIF)/CD74 signaling pathway, thus reducing fibrogenesis and liver damage [130]. Additionally, VA has been investigated in HepG2 cells to explore its effect on glucokinase (GK), a crucial enzyme in hepatic glucose metabolism and a glucose sensor in the liver. In the study by Sreelekshmi and Raghu, VA was shown to activate GK through the BAD pathway. This activation has significant potential for developing future therapies targeting diabetes, as enhancing GK activity can improve glucose regulation and metabolic control [132].

#### 3.2.6. Sinapic Acid (SA)

SA, also known as 4-hydroxy-3,5-dimethoxycinnamic acid, is a phenylpropanoid compound with well-documented antioxidant, anti-inflammatory, and neuroprotective properties. Recently, its hepatoprotective effects have gained attention (Table 2). In a study by Shin et al., SA demonstrated significant hepatoprotective and antifibrotic effects in rats with DMN-induced chronic liver injury [133]. These protective effects were attributed to SA’s potent antioxidant activity in scavenging free radicals, its ability to suppress TGF-β1, and its capacity to inhibit the activation of hepatic stellate cells, which are key drivers in liver fibrosis progression [133].

Other studies have highlighted the potential of SA in treating drug-induced liver injury. For instance, Ahmad et al. reported that SA mitigated MTX-induced liver injury in rats by inhibiting apoptosis and enhancing the expression of antioxidant enzymes through the nuclear factor erythroid 2-related factor 2/heme oxygenase-1 (Nrf2/HO-1) pathway, activated by the inhibition of NF-κB [134]. This mechanism highlights SA’s potential as a therapeutic agent for patients undergoing long-term MTX treatment, a potent antimetabolite and immunosuppressant used to treat neoplastic and autoimmune disorders, by reducing MTX-induced liver toxicity [134].

Additionally, SA’s ability to ameliorate liver injury was further demonstrated in a study by Rostami et al., particularly in cases of paracetamol (acetaminophen; APAP)-induced acute liver injury [135]. In this study, supplementation with 50 mg/kg of SA protected the liver from the adverse hepatotoxic effects of APAP. This protective effect was linked to the modulation of the NF-κB/Nrf2/HO-1 signaling pathway, as well as the regulation of sirtuin 1 (Sirt1), mitochondrial integrity, and lysosomal stabilization [135].

#### 3.2.7. Syringic Acid (SYRA)

Among phenolic compounds, SYRA, also known as 4-hydroxy-3,5-dimethoxybenzoic acid, is gaining recognition as an emerging nutraceutical, with a history of use in traditional Chinese medicine [136]. Its hepatoprotective effects have recently been explored in animal models, while its potential anticancer properties have drawn attention (Table 2). In a study by Gheena and Ezhilarasan, the cytotoxic effects of SYRA on hepatocellular carcinoma were evaluated using HepG2 cell models [137]. The results showed that SYRA treatment induced significant cytotoxicity and ROS production in HepG2 cells, which, in turn, activated apoptotic pathways. This was evidenced by increased expression of apoptotic markers such as caspases 3 and 9, cytochrome c, Apaf-1, Bax, and p53 genes, indicating that SYRA may promote apoptosis in liver cancer cells [137].

The anti-inflammatory effects of SYRA on the liver have been demonstrated in recent studies. Somade et al. showed that SYRA modulates the nuclear factor kappa B-inducible nitric oxide synthase -cyclooxygenase 2 (NF-κB-iNOS-COX-2) and JAK-STAT pathways in methyl cellosolve (MCEL)-induced hepato-testicular inflammation [138]. SYRA treatment at 50 and 75 mg/kg significantly reduced levels of IL-6, TNF-α, iNOS, COX-2, and NF-κB, while also decreasing JAK1 and SOCS1 mRNA expression in the liver. These findings indicate that SYRA inhibits MCEL-induced inflammation by suppressing the NF-κB and JAK-STAT signaling pathways [138].

Other studies have also highlighted the potential of SYRA in treating liver injury. In the study by Ferah Okkay et al., the effects of SYRA on thioacetamide-induced hepatic encephalopathy, a serious syndrome associated with neuropsychiatric abnormalities due to acute liver dysfunction, such as cirrhosis, were evaluated [139]. SYRA significantly alleviated thioacetamide-induced hepatic injury by reducing hepatotoxicity biomarkers, including ammonia, AST, ALT, ALP, and lactate dehydrogenase (LDH) levels, while also decreasing oxidative stress (lower MDA, ROS) and increasing antioxidant activity (higher SOD and GSH). Additionally, SYRA attenuated inflammatory damage by suppressing pro-inflammatory cytokines TNF-α, IL-1β, and NF-κB and promoting anti-inflammatory interleukin-10 (IL-10) [139]. Another study by Adeyi et al. further demonstrated the potential of SYRA in treating DMN-induced hepatotoxicity in rats [140]. SYRA treatment significantly reduced the activities of liver enzymes (ALT, AST), endogenous antioxidants (GPx, CAT, SOD), and markers of oxidative stress (MDA, GSH). Additionally, it lowered pro-inflammatory cytokines (TNF-α, IL-1β) and NF-κB levels. These effects were achieved through SYRA’s ability to maintain endogenous antioxidant defenses while downregulating pro-inflammatory and apoptotic markers, highlighting its hepatoprotective properties [140].

#### 3.2.8. Chlorogenic Acid (CGA)

CGA, a phenolic secondary metabolite formed by the esterification of caffeic acid and quinic acid, has garnered attention for its hepatoprotective properties (Table 2) [141]. Zhu et al. explored CGA’s potential to mitigate alcoholic liver injury in mice, showing significant reductions in serum ALT, AST, low-density lipoprotein (LDL) cholesterol, total cholesterol (TC), and triglyceride (TG) levels in ethanol-fed mice [142]. Additionally, CGA restored gut microbial balance by increasing beneficial bacteria, suggesting its hepatoprotective effects are mediated through the modulation of gut–liver axis homeostasis. These findings highlight CGA’s therapeutic potential in liver health [142].

Miao et al. further demonstrated the potential of CGA in alleviating liver fibrosis in a methionine- and choline-deficient (MCD) diet MASH mouse model [143]. CGA inhibited the activation of HSCs and promoted mitochondrial biogenesis both *in vivo* and *in vitro* [143]. Additionally, CGA reduced hepatic ECM accumulation and serum high-mobility group box 1 (HMGB1) levels. These results suggest that CGA mitigates liver fibrosis by inhibiting HSC activation, enhancing mitochondrial biogenesis, and reducing HMGB1-triggered ECM production in hepatic endothelial cells [143].

#### 3.2.9. Luteolin (LU)

LU (3′,4′,5,7-tetrahydroxyflavone), a naturally occurring flavonoid known for its anticancer properties, has garnered interest for its hepatoprotective potential, particularly in fatty liver diseases (Table 2) [144]. El-Asfar et al. demonstrated LU’s ability to alleviate tamoxifen (TAM)-induced fatty liver in rats, showing reduced hepatic steatosis, decreased serum aminotransferase levels, and lowered hypertriglyceridemia [145]. Additionally, LU suppressed hepatic inflammation and increased hepatic β-catenin, suggesting a protective effect in TAM-induced fatty liver [145]. Similarly, a study by Liu et al. highlighted LU’s efficacy in mitigating MASLD [146]. The compound inhibited the Toll-like receptor 4 (TLR4) signaling pathway in the liver, reducing pro-inflammatory cytokine secretion and alleviating MASLD symptoms [146]. Furthermore, LU significantly altered gut microbiota composition in MASLD rats, enhancing microbial richness, which may contribute to its hepatoprotective effects [146].

Several studies have highlighted LU’s potential in treating liver injury. Wang et al. demonstrated that LU ameliorates lipopolysaccharide (LPS)-induced acute liver injury in mice by inhibiting the thioredoxin-interacting protein/nucleotide oligomerization domain (TXNIP/NOD)-like receptor protein 3 (NLRP3) inflammasome [147]. LU also reduced liver inflammation by suppressing pro-inflammatory genes, including TNF-α, IL-10, and IL-6. Additionally, it alleviated LPS-induced hepatocyte injury by reducing oxidative stress and modulating MDA, SOD, and GSH levels [147].

A recent study by Batudeligen et al. demonstrated LU’s strong protective effects against liver fibrosis in rat hepatic stellate cells (HSC-T6) [148]. Western blot analysis revealed that proteins C-C chemokine receptor type 1 (CCR1), CD59, and alpha-N-acetylgalactosaminidase (NAGA), which were downregulated by TGF-β1 treatment (a fibrosis inducer), were upregulated with LU treatment [148]. Additionally, proteins such as inter-alpha-trypsin inhibitor heavy chain H3 (ITIH3), the marker of proliferation Ki-67 (MKI67), kinesin-like protein KIF23, and others, which were elevated in response to TGF-β1, were reduced by LU [148]. This suggests LU’s potential to counteract fibrosis-related protein changes in liver cells [148].

Recent research suggests LU may have potential in liver transplantation treatment. In a study by Wang et al., LU was found to attenuate acute liver allograft rejection in rats [149]. While no human studies have yet confirmed this, the findings showed that LU significantly protected liver graft structure and function, prolonged recipient survival, reduced T cell infiltration, and downregulated pro-inflammatory cytokines [149]. Additionally, LU inhibited the proliferation of CD4^+^ T cells, indicating its potential in managing immune response during liver transplantation [149].

#### 3.2.10. Apigenin (AP)

AP (4′,5,7-trihydroxyflavone), a well-known phenolic compound with various nutritional and organoleptic properties, has recently been studied for its health benefits, particularly its hepatoprotective effects (Table 2) [150]. In a study by Zhao et al., AP was shown to prevent acetaminophen-induced liver injury in mice through the activation of the Sirt1 pathway [151]. Treatment with AP (80 mg/kg) significantly reduced levels of liver damage markers such as ALT/AST, MDA, liver myeloperoxidase (MPO) activity, GSH, and ROS. Additionally, AP promoted Sirt1 expression and deacetylated p53, further highlighting its protective role [151].

A recent study by Ji et al. further highlighted the potential of AP in alleviating liver fibrosis [152]. AP was demonstrated to reduce ALT and AST levels, suppress ECM production, and inhibit HSC activation [152]. Additionally, it regulated the balance between MMP-2 and TIMP-1, reduced the expression of autophagy-related proteins, and inhibited the TGF-β1/Smad3 and p38 MAPK/peroxisome proliferator-activated receptor α (p38/PPARα) signaling pathways. These findings suggest AP’s promising role in preventing liver fibrosis [152].

A study by Meng et al. revealed that AP alleviates high-fat diet-induced liver injury by reducing pyroptosis through the mitophagy-reactive oxygen species-cathepsin B-NOD-like receptor pyrin domain-containing 3 (mitophagy-ROS-CTSB-NLRP3) pathway [153]. AP lowered inflammatory markers and oxidative stress, promoted mitophagy, and stabilized lysosomal membranes, preventing liver cell death [153].

#### 3.2.11. Other Compounds

While several compounds have been recently explored for their hepatoprotective effects, others such as γ-O, EP, RU, QU, and VX have limited available data regarding their hepatoprotective properties (Table 2). Further research is needed to thoroughly investigate their potential mechanisms and efficacy in liver protection.

γ-O, a mixture of ferulic acid esters of triterpene alcohols and phytosterols, has been shown to protect human liver cells (L02) from hydrogen peroxide (H_2_O_2_)-induced oxidative damage, as reported by Ma et al. [154]. The protective effect of γ-O is mediated through the MAPK/Nrf2 signaling pathways, where preincubation with γ-O reduced H_2_O_2_-induced expression of the pro-apoptotic protein BAK (Bcl-2 antagonist/killer) and the phosphorylation of apoptosis signal-regulating kinase 1 (ASK1), p38, c-Jun NH_2_-terminal kinase (JNK), and extracellular signal-regulated kinase (ERK) [154]. Additionally, γ-O increased the expression of the anti-apoptotic protein Bcl-xl and antioxidant proteins Nrf2 and HO-1 [154].

EP, a plant-derived secondary metabolite, has exhibited protective effects against amebic liver abscesses in hamsters in the study by Velásquez-Torres et al. [155]. It promoted liver repair, inhibited amoebae presence, and modulated inflammatory cytokines (TNF-α, IL-1β, IL-10), reducing liver abscess progression to just 9.49%, compared to 84% in untreated animals [155].

RU (3-rhamnosyl-glucosyl quercetin) was recently reported by Naderi et al. to mitigate perfluorooctanoic acid (PFOA)-induced liver injury by modulating oxidative stress, apoptosis, and inflammation [156]. Rutin decreased the BAX:Bcl-2 ratio and reduced PFOA-induced gene expression of TNF-α, IL-6, NF-κB, and JNK [156]. These findings suggest that rutin may hold potential for preventing and treating hepatotoxicity caused by environmental toxins [156].

QU proved to ameliorate MASLD by promoting AMPK-mediated hepatic mitophagy [157]. It effectively reduced intracellular lipids, ROS, and inflammation, both *in vivo* and *in vitro*, but these effects were blocked by 3-MA (3-methyladenine, an autophagy inhibitor) and CC (Compound C, an AMPK inhibitor) [157]. This study highlights quercetin’s potential as a therapeutic agent for MASLD by enhancing mitophagy through AMPK activation.

VX has been shown to protect against ethanol-induced liver injury by modulating the Sirt1/p53 signaling pathway. *In vitro*, VX restored cell activity and reduced AST release in LO2 cells, while *in vivo*, it decreased aminotransferase, blood lipid, and uric acid levels, and improved liver pathology [158]. Additionally, vitexin restored Sirt1/Bcl-2 expression and inhibited caspase-3, p53, and ac-p53 elevation [158].

**Table 2 nutrients-17-00900-t002:** Effects of several enhanced phenolic components of fermented cereals to liver health.

Compounds	Liver Diseases	Pathway/Mechanism of Action	Main Biomarkers	Object	References
*trans-*Ferulic acid (*t*FA)	Liver fibrosis	Inhibited transforming growth factor β1 (TGF-β1) pathway	- p-Smad2 (phosphorylated Smad2) (Decreased) - p-Smad3 (phosphorylated Smad3) (Decreased) - Smad4 (Decreased)	*In vivo*: Sprague−Dawley rats *In vitro*: human hepatic stellate cell line (HSC) LX-2	[105]
	Methotrexate (MTX)-induced liver damage	Improved antioxidant levels and decreased inflammatory markers	- Nitric oxide (NO) (Decreased) - Malondialdehyde (MDA) (Decreased) - Interleukin-6 (IL-6) (Decreased) - Tumor necrosis factor-α (TNF-α) (Decreased) - Reduced glutathione (GSH) (Increased)	*In vivo*: 8-week-old male Swiss albino mice (22–25 g)	[87]
	Liver fibrosis	Inhibited PTP1B activity and promoted AMPK phosphorylation	- p-AMPK (Phosphorylated AMPK) (Increased) - PTP1B (Protein Tyrosine Phosphatase 1B) (Inhibited) - ALT (Alanine aminotransferase) (Decreased) - AST (Aspartate aminotransferase) (Decreased) - HSC (hepatic stellate cell) (Inhibited)	*In vivo*: male and female C57BL/6J mice (22–24 g)	[106]
	Acetaminophen (APAP)-induced acute liver injury	AMPK-mediated antioxidant, anti-apoptosis: Hepatoprotection	- ALT (Decreased) - AST (Decreased) - HNF4a (Hepatocyte nuclear factor 4 alpha) (Upregulated) - BAX (Bcl-2-associated X protein) (Decreased)	*In vivo*: male and female C57BL/6J mice (22–24 g)	[88]
	Iron-induced liver injury	Erythroid-2-related factor 2 (Nrf2)-mediated antioxidant gene activation: Anti-iron oxidative stress	- ROS (Reactive oxygen species) (Decreased) - Lipid peroxidation (Decreased) - Superoxide dismutase (SOD) (Increased) - Glutathione (GSH) (Increased)	*In vivo*: 6-week-old BALB/c male mice (24–30 g) *In vitro*: hepatocellular carcinoma (HepG2) cells	[108]
	Gamma radiation- induced liver injury	JAK/STAT/Nrf2 pathway: Reduced Oxidative Stress/Ferroptosis	- JAK/STAT signaling pathway (Inhibited) - Nrf2 signaling pathway (Upregulated) - GPX4 (Glutathione peroxidase 4) (Increased) - SLC7A11 (Solute carrier family 7 member 11) (Increased)	*In vivo*: 24 adult male rats	[94]
*p*-Coumaric acid (*p*CA)	Acetaminophen (AAP)-induced liver injury	Reduced liver enzymes and suppressed apoptosis via ROS and mitogen-activated protein kinase (MAPK) signaling	- ALT (Decreased) - AST (Decreased) - ROS (Decreased) - MAPK (Regulated)	*In vivo*: 8-week-old male C57BL/6 mice	[109]
	Fipronil-induced liver injury	Regulated antioxidant enzymes and reduced inflammatory cytokines	- ALT (Decreased) - AST (Decreased) - TNF-α (Decreased) - SOD (Increased) - Lipid peroxidation (Decreased)	*In vivo*: 30 male Swiss albino mice (4–6 weeks age, 25–28 g body weight)	[89]
	Radiation-induced liver damage (RILD)	Reduced liver enzymes, inhibited BCL2-associated X protein (BAX) expression, and improved hematopoietic function	- ALT (Decreased) - AST (Decreased) - BAX (Inhibited)	*In vivo*: C57BL/6 male mice	[112]
	Liver damage induced by dust and ischemia-reperfusion (IR) injury	Protection against oxidative stress, inflammation, and autophagy	- ALT (Decreased) - AST (Decreased) - MDA (Decreased) - TNF-α (Decreased) - NF-κB (Nuclear factor kappa-light-chain-enhancer of activated B cells) (Decreased) - HOTAIR (HOX transcript antisense intergenic RNA) (Decreased) - HOX-1 (Homeobox Gene 1) (Decreased)	*In vivo*: 48 adult male Wistar rats	[93]
Gallic acid (GA)	Paracetamol-induced liver damage	Increased antioxidant activity, reduced liver marker enzyme levels and inflammatory mediators	- TNF-α (Decreased) - ALT (Decreased) - AST (Decreased)	*In vivo*: male crossbreed Swiss albino mice (20–25 g)	[115]
	CCl_4_-mediated liver fibrosis	Reduced oxidative stress, liver damage markers, and inhibited hepatic stellate cell activity via MMP-2/TIMP-1 regulation	- ALT (Decreased) - AST (Decreased) - γ-GT (Gamma-Glutamyl Transferase) (Decreased) - MDA (Decreased) - HA (Hyaluronic acid) (Decreased) - CIV (Type IV collagen) (Decreased) - MMP-2 (Matrix metalloproteinase-2) (Decreased) - TIMP-1 (Tissue inhibitor of metalloproteinases-1) (Decreased)	*In vivo*: BALB/c mice (18–22 g)	[120]
	Hepatic ischemia and reperfusion (IR) injury	Reduced oxidative stress, increased antioxidant enzymes, reduced liver enzymes: restored antioxidant balance	- MDA (Decreased) - CAT (Increased) - GPx (Increased) - ALT (Decreased) - AST (Decreased) - LDH (Lactate dehydrogenase) (Decreased)	*In vivo*: 32 adult Spraque–Dawley albino rats	[93]
	Toxicity induced by Pb in liver	Reduced oxidative damage and enhanced antioxidant defenses	- ALA-D (Aminolevulinic acid dehydratase) (Increased) - Lipid peroxidation (Decreased) - Protein oxidation (Decreased) - AST (Decreased) - SOD (Increased) - CAT (Increased) - GSH (Increased)	*In vivo*: 32 about 2-month-old male Wistar rats (240 ± 10 g)	[122]
	Dimethylnitrosamine (DMN)-induced liver fibrosis	Reduced DMN-induced liver fibrosis: antioxidant, reduced pro-fibrotic cytokines, liver/kidney protection	- ALT (Decreased) - AST (Decreased) - Cytokines (Decreased) - Antioxidant capacity (Increased)	*In vivo*: 48 male Sprague–Dawley rats (180–200 g)	[121]
	Ketamine-induced oxidative damage	Reduced protein carbonyl (PC) levels and increased non-protein thiol (NPSH) levels	- RS (Reactive species) (Decreased) - PC (Decreased) - NPSH (Increased)	*In vivo*: 32 Wistar male rats (150–200 g)	[117]
	Diclofenac-induced liver toxicity	Enhanced antioxidant defenses, reduced oxidative stress, inflammation, and liver damage markers	- Antioxidant defenses: GSH, GPx (Glutathione peroxidase), SOD, CAT (Catalase) (Increased) - ALT (Decreased) - AST (Decreased) - ALP (Alkaline phosphatase) (Decreased) - MDA (Decreased) - IL-1β (Interleukin-1 beta) (Decreased)	*In vivo*: 30 male 6–8-week-old Wistar rats (200 ± 20 g)	[118]
	Isoniazid and rifampicin-induced liver injury	Enhanced antioxidant defense (Nrf2 activation), inhibited inflammation (NF-κB/TLR-4 pathway)	- Nrf2 (Increased) - NF-κB (Nuclear factor Kappa B) (Inhibited) - TLR-4 (Toll-like receptor 4) (Decreased) - ALT (Decreased) - AST (Decreased)	*In vivo*: 4–6 months old male Wistar rats (200–250 g)	[119]
	Dust-induced MASLD	Inhibited oxidative stress/inflammation (antioxidant), protected liver function, but had a limited lipid profile effect	- ALT (Decreased) - AST (Decreased) - MDA (Decreased) - TAC (Total Antioxidant Capacity) (Increased) - NF-κB (Decreased) - TNFα (Decreased) - HO1 (Heme oxygenase-1) (Increased) - miR-122 and miR-34a (Decreased)	*In vivo*: 24 adult male Wistar rats (200–250 g)	[91]
	Type 2-induced diabetic liver injury	Modulated glucose/insulin/liver enzymes, restored redox balance (GLUT-4/Wnt1/β-catenin upregulation), inhibited ERK1/2-NF-κB (inflammation), increased GLP-1 (insulin sensitivity)	- GLUT-4 (Glucose transporter type 4) (Upregulated) - Wnt1/β-catenin pathway (Activated) - ERK1/2-NF-κB pathway (Inhibited) - GLP-1 (Glucagon-like peptide-1) (Increased) - Fetuin-A (Decreased) - Liver enzymes (ALT/AST) (Reduced)	*In vivo*: 60 adult (3–4 months old) male healthy Sprague–Dawley rats (180 ± 20 g)	[124]
	MASLD	Activated AMPK-ACC-PPARα axis (lipid metabolism, mitochondrial function, oxidative stress)	- AMPK (AMP-activated protein kinase) (Activated) - ACC (Acetyl-CoA carboxylase) (Inactivated by AMPK, reducing fat accumulation) - PPARα (Peroxisome proliferator-activated receptor alpha) (Enhanced) - mtROS (Mitochondrial reactive oxygen species) (Reduced) - Liver enzymes (ALT/AST) (Reduced) - Lipid deposition (Decreased)	*In vitro*: HepG2 and SMMC-7721 cells	[69]
	Sepsis-induced liver injury	Inhibited MAPK, reduced ERK1/2/NF-κB inflammation, enhanced antioxidant defenses	- MAPK signaling (Suppressed) - ERK1/2 (Inhibited) - NF-κB (Downregulated) - C/EBPβ (Involved in modulating inflammatory pathways, affected by GA’s regulation) - Oxidative stress markers (Improved by GA) - Liver enzymes (ALT/AST) (Lowered)	*In vivo*: mice (Not specifically mentioned)	[125]
Caffeic acid (CA)	Liver reperfusion injury	Upregulated Sirt3, enhanced mitochondrial respiratory chain (MRC) activity, reduced oxidative stress/microcirculatory disturbances	- Sirt3 expression (Inhibited) - MRC activity (mitochondrial respiratory chain) (Upregulated) - NADH dehydrogenase (ubiquinone) 1 alpha subcomplex subunit 9 (acetylation level) (Decreased) - Succinate dehydrogenase complex subunit A (acetylation level) (Decreased)	*In vivo*: male Sprague–Dawley rats (200–720 g)	[127]
	Liver injury after transplantation	Inhibited PDIA3, reduced NADPH oxidase activity/ROS production, prevented apoptosis, improved liver function/reduced microcirculatory damage	- PDIA3 (protein disulfide isomerase A3) expression (Attenuated) - NADPH oxidase activity (Attenuated) - ROS production (Decreased) - Liver function markers (ALT, AST, etc.) (Attenuated) - Apoptosis (cell death rate) (Reduced)	*In vivo*: male Sprague–Dawley rats (210 ± 20 g)	[105]
	Non-alcoholic fatty liver disease induced by a high-fat diet	Reversed HFD-induced gut microbiota dysbiosis (reduced LPS inflammation, regulated lipid metabolism genes)	- Lipid accumulation (Decreased) - Serum biochemical parameters (Decreased) - Inflammatory markers (Decreased) - Gene expression (Regulated)	*In vivo*: 6-week-old male C57BL/6J mice	[129]
Vanillic acid (VA)	Diethylnitrosamine and 1,2-Dimethylhydrazine-induced hepatocarcinogenesis	Induced detoxification enzymes (GSTA-5, Nrf-2), reduced proliferation (Cyclin D1), promoted apoptosis (Caspase-3/BAD upregulation, Bcl-2 downregulation)	- Carcinogenesis markers: Hepatic glutathione S-transferase placental form (GST-P) positive foci (Reduced) - Cell proliferation marker: proliferating cell nuclear antigen (PCNA), Cyclin D1 (Decreased) - Apoptosis Markers: Caspase-3 (Increased), BAD (Increased), Bcl-2 (Decreased) - Detoxification markers: GSTA-5, Nrf-2 (Increased)	*In vivo*: rats (Not mentioned specifically)	[131]
	Impairments in glucose metabolism in HepG2 cells	Activated glucokinase (GK), prevented glycogen depletion (hyperinsulinemia, HepG2 cells), activated Bcl-2-associated death receptor (BAD) (molecular docking: VA-GK interaction)	- GK (Glucokinase) (Safeguarded) - BAD (Bcl-2-associated death promoter) (Increased) - PEPCK (Phosphoenolpyruvate Carboxykinase) (Reduced) - G6Pase (Glucose-6-Phosphatase) (Reduced) - GSK-3β (Glycogen synthase kinase-3β) (Normalized)	*In vitro*: HepG2 cells	[132]
	CCl_4_-induced liver fibrosis	Inhibited HSC activation/proliferation/migration (MIF/CD74 pathway, autophagy inhibition)	- Fibrosis markers: MIF (Macrophage migration inhibitory factor), CD74 (Cluster of differentiation 74), α-SMA (Alpha-smooth muscle actin), CIV (Reduced) - Autophagy marker: LC3B (Microtubule-associated protein 1A/1B-light chain 3B) (Reduced) - HSC Activity Indicators: Proliferation and migration of HSCs (Inhibited)	*In vivo*: 40 male 8-week-old Sprague–Dawley rats (200–220 g)	[130]
Sinapic acid (SA)	Dimethylnitrosamine (DMN)-induced hepatic fibrosis	Exerted hepatoprotective/antifibrotic effects (antioxidant, scavenged free radicals, downregulated TGF-β1/HSC activation)	- Serum enzymes: ALT and AST (Decreased) - Liver lipid peroxidation: MDA (Decreased) - Fibrosis markers: Hydroxyproline, CIV mRNA, α-smooth muscle actin (α-SMA), TGF-β1 (Decreased)	*In vivo*: male Sprague–Dawley rats	[133]
	Methotrexate (MTX)-induced hepatic injuries	Modulated Nrf2/HO-1/NF-κB pathways, inhibited apoptosis, enhanced antioxidant enzyme activity	- Liver function indices: ALT, AST, ALP (Decreased) - Antioxidant defenses: GSH, SOD, CAT (Increased) - Oxidative stress markers: MDA, NO (Decreased) - Inflammatory cytokines: TNF-α, IL-1β, myeloperoxidase (MPO) (Decreased)	*In vivo*: 6-week-old male Wistar rats (250 ± 10 g)	[134]
	Paracetamol-induced acute liver injury	Protected against paracetamol-induced liver injury (antioxidant/anti-inflammatory, modulated NF-κB/Nrf2/HO-1, regulated sirtuin 1, maintained mitochondrial/lysosomal integrity)	- Liver function markers: ALT, ALP, AST (Decreased) - Oxidative stress markers: ROS, MDA (Decreased) - Inflammatory cytokines: IL-6, TNF-α, MPO, NF-κB (Decreased) - Antioxidant proteins: Sirtuin 1, Hemoxygenase-1 (HO-1), Nrf2, SOD (Increased) - Lysosomal enzymes: Cathepsin B, β-galactosidase (Decreased)	*In vivo*: male C57BL/6 mice	[135]
Syringic acid (SYRA)	Hepatocellular carcinoma	Induced HepG2 cytotoxicity (apoptosis: upregulated caspases/cytochrome c/Apaf-1/BAX/p53, downregulated Bcl-2, increased ROS)	- Cytotoxicity indicators: 3-(4,5-dimethylthiazol-2-yl)-2,5-diphenyltetrazolium bromide (MTT) assay (Induced) - ROS (measured by dichlorofluorescein staining) (Increased) - Apoptotic markers: Upregulated (Caspases 3 and 9, Cytochrome c, Apaf-1, BAX, p53), Downregulated (Bcl-2)	*In vitro*: HepG2 cell line	[137]
	Thioacetamide-induced hepatic encephalopathy	Decreased liver dysfunction enzymes/oxidative markers, promoted antioxidant activity, suppressed inflammatory cytokines, improved liver health (downregulated inflammation/apoptosis/stress markers)	- Liver function markers: Ammonia, AST, ALT, ALP, LDH (reduced by SYRA) - Oxidative stress markers: MDA, ROS (Reduced), SOD, GSH (Increased) - Inflammatory markers: TNF-α, IL-1β, NF-κB (reduced), IL-10 (Increased) - Apoptosis markers: Caspase-3 (Downregulated)	*In vivo*: 24 adult male Wistar rats (260–320 g)	[139]
	Methyl cellosolve-induced hepato-testicular inflammation	Inhibited NF-κB/JAK-STAT pathways, decreased pro-inflammatory markers (reduced JAK1/STAT1/SOCS1 activation), downregulated PIAS1 (liver/testis)	- Pro-inflammatory markers: IL-6, TNF-α, iNOS, COX-2, NF-κB (Decreased) - JAK-STAT pathway: JAK1 and STAT1 (Downregulated by SYRA) - Protein inhibitor of activated STAT1 (PIAS1) (Reduced by SYRA)	*In vivo*: 30 male Wistar rats (≅ 220 g)	[138]
	DMN-induced hepatotoxicity	Reducing oxidative stress and inflammation	- Liver function markers: ALT, AST (Reduced by SYRA) - Oxidative stress markers: MDA, NO (Reduced by SYRA), GSH, GPx, CAT, SOD (Increased by SYRA) - Inflammatory markers: TNF-α, IL-1β, NF-κB (Reduced by SYRA)	*In vivo*: 30 male Wistar rats (≅ 200 g)	[140]
Chlorogenic acid (CGA)	Alcohol liver injury	Reduced liver damage, promoted gut health (increased beneficial bacteria/n-butyric acid, supported intestinal barrier), decreased liver injury inflammatory markers	- Liver function markers: ALT, AST (Reduced by CGA) - Lipid profile markers: LDL (Low-density lipoprotein), TC (Total cholesterol), TG (Triglycerides) (Reduced by CGA) - Increased beneficial bacteria: *Muribaculaceae*, *Bacteroides*, *Alloprevotella*, *Parabacteroides* - Decreased harmful bacteria: *Eubacterium nodatum*, *Eubacterium ruminantium*, *Anaerotruncus* - Short-chain fatty acids: n-butyric acid (Increased by CGA) - Inflammatory cytokines (Reduced by CGA)	*In vivo*: 30 male Kunming mice (20 ± 2 g)	[142]
	Liver fibrosis in methionine and choline-deficient diet-induced nonalcoholic steatohepatitis	Reduced liver fibrosis (inhibited HSC activation, promoted mitochondrial biogenesis, decreased HMGB1-induced ECM production)	- PGC1α (Peroxisome proliferator-activated receptor gamma coactivator 1-alpha, linked to mitochondrial biogenesis) (Increased) - HMGB1 (High-mobility group box 1) (Decreased) - Collagen (Fibrosis marker) (Decreased) - ECM (Extracellular matrix) (Decreased)	*In vivo*: specific pathogen-free male C57BL/6 mice (20 ± 2 g)	[143]
Luteolin (LU)	Tamoxifen (TAM)-associated fatty liver	Protected against TAM-induced cognitive impairment/liver steatosis (reduced inflammation, modulated hepatic β-catenin)	- β-catenin (Increased) - Serum aminotransferases (Decreased) - Triglycerides (Decreased) - Hepatic inflammatory markers (Decreased)	*In vivo*: 8-week-old male albino rats (150–200 g)	[145]
	Non-alcoholic fatty liver disease in rats	Restored intestinal barrier integrity, modulated gut microbiota, inhibited TLR4 signaling (reduced liver inflammation)	- Tight junction proteins (Increased) - TLR4 (Toll-like receptor 4) (Inhibited) - Pro-inflammatory factors (Decreased) - Gut microbiota diversity (Increased) - LPS (Lipopolysaccharides) levels (Decreased)	*In vivo*: 48 6-week-old Wistar rats	[146]
	LPS-induced acute liver injury	Inhibited TXNIP-sNLRP3 axis, reduced inflammation/oxidative stress/apoptosis	- TXNIP (Thioredoxin-interacting protein) (Decreased) - NLRP3 (Inhibited) - ASC (Apoptosis-associated speck-like protein) (Decreased) - Caspase-1, IL-1β, IL-18 (Decreased) - TNF-α, IL-6, IL-10 (Decreased) - MDA (Decreased) - SOD, GSH (Increased)	*In vivo*: male C57BL/6 mice	[147]
	Liver fibrosis	Modulated fibrosis-related gene expression, regulated DNA replication/lysosomal signaling/collagen biosynthesis proteins, promoted fibrosis reversal enzyme activity/cellular functions	- CCR1, CD59, NAGA (Fibrosis-promoting markers) (Decreased) - ITIH3, MKI67, KIF23, DNMT1, P4HA3, CCDC80, APOB, FBLN2 (Fibrosis-protective markers) (Increased) - TGFβ1 (Fibrosis inducer) (Inhibited) - Collagen (Fibrosis marker) (Decreased)	*In vitro*: HSC-T6 cells	[148]
	Acute liver allograft rejection in rats	Inhibited CD4+ T cell proliferation/Th1 differentiation, promoted regulatory T cells (Tregs) (enhanced immunosuppression, reduced pro-inflammatory cytokines)	- Pro-inflammatory cytokines (e.g., IL-2, IFN-γ) (Decreased) - CD4+ T cells (Th1 cells) (Decreased) - Regulatory T cells (Tregs) (Increased) - T cell infiltration in liver grafts (Ameliorated)	*In vivo*: 8–10-week-old male Lewis rats and Brown Norway rats (200–250 g)	[149]
Apigenin (AP)	Acetaminophen- induced liver injury	Regulated SIRT1-p53 axis, promoted autophagy, activated NRF2 pathway, inhibited nuclear p65 activation (alleviated inflammation/oxidative stress)	- ALT, AST (Decreased) - MDA (Decreased) - Liver myeloperoxidase (MPO) activity (Decreased) - GSH (Increased) - ROS (Decreased) - SIRT1 expression (Increased) - p53 acetylation (Decreased)	*In vivo*: C57BL/6 mice (20 ± 2 g)	[151]
	Liver fibrosis	Inhibited HSC activation/autophagy (TGF-β1/Smad3, p38/PPARα pathways)	- ALT (Decreased) - AST (Decreased) - ECM (Inhibited) - HSC (Activation inhibited) - MMP2 (Matrix metalloproteinase 2) (Regulated) - TIMP1 (Tissue inhibitor of metalloproteinases 1) (Regulated) - Autophagy-related proteins (Decreased expression)	*In vivo*: 66 male 6-week-old C57 mice (22–26 g)	[152]
	High-fat diet-induced hepatic pyroptosis	Mitophagy-ROS-CTSB-NLRP3 pathway	- LDH (Lactate dehydrogenase) (Decreased) - NLRP3 (NOD-like receptor family pyrin domain-containing 3) (Decreased) - GSDMD-N (*N*-terminal domain of Gasdermin D) (Decreased) - Cleaved-caspase 1 (Decreased) - CTSB (Cathepsin B) (Decreased) - IL-1β (Decreased) - IL-18 (Decreased) - LAMP-1 (Lysosomal-associated membrane protein-1) (Increased)	*In vivo*: 30 mice	[153]
γ-Oryzanol (γ-O)	Hydrogen peroxide-induced oxidative damage	Inhibited MAPK signaling, activated Nrf2 signaling	- Hydroxyl radicals (OH·) (Decreased) - NO (Decreased) - Pro-apoptotic protein BAK (Decreased) - Phosphorylation of ASK1, p38, JNK, and ERK (Decreased) - Anti-apoptotic protein Bcl-xl (Increased) - Antioxidative stress proteins Nrf2 and HO-1 (Increased)	*In vitro*: human hepatic cells (L02 cells)	[154]
Epicatechin (EP)	Amebic liver abscess (ALA) development in hamsters	Promoted liver repair, modulated inflammatory responses (eliminated *E. histolytica*, aided liver healing)	- Liver abscess damage: (Decreased) (9.49% damage vs. 84% in untreated) - Inflammatory cytokines: TNF-α (Decreased), IL-1β (Decreased), IL-10 (Increased)	*In vivo*: male 2-month-old Syrian golden hamsters (*Mesocricetus auratus*) (100 ± 5 g)	[155]
Rutin (RU)	Perfluorooctanoic acid (PFOA)-induced liver	Exerted protective effects against PFOA-induced liver injury (inhibited oxidative stress, alleviated mitochondrial dysfunction, reduced inflammation)	- Liver enzymes (Mitigated by rutin treatment) - Histopathological defects (Ameliorated) - Mitochondrial dysfunction (Reduced) - BAX: Bcl2 ratio (Decreased) - TNF-α (Reduced) - IL-6 (Reduced) - NF-κB (Reduced) - JNK (Reduced)	*In vivo*: male Wistar rats (200–250 g)	[156]
Quercetin (QU)	Nonalcoholic fatty liver disease (MASLD)	Alleviated MASLD (AMPK-mediated mitophagy, enhanced damaged mitochondria removal, anti-inflammatory effects)	- Serum ALT/AST (Decreased) - AMPK (Increased activation, suggesting enhanced autophagy and mitophagy)	*In vivo*: male 42-day-old C57BL/6J (18–22 g)	[157]
Vitexin (VX)	Ethanol-induced liver injury	Sirt1/p53-mediated mitochondrial apoptotic pathway: promoted cell survival, inhibited apoptosis	- AST/ALT (Decreased) - Sirt1/Bcl-2 (Increased expression) - Caspase-3 (Decreased) - MDA: Reduced - TNF-α: Decreased	*In vivo*: male Kunming mice (18–22 g)	[158]

## 4. Effects of Fermented Cereals on Liver Health (*In Vitro* & *In Vivo*)

Research on the effects of fermented cereals on liver health is growing, with recent studies exploring their broader impact beyond individual phenolic compounds. While human studies are lacking, *in vitro* and *in vivo* models have provided valuable insights, primarily focusing on cereals such as barley, rice bran, oats, and wheat. Expanding research to a wider range of cereals is crucial.

### 4.1. Effects on Lipid Metabolism and Steatosis

A significant aspect of liver health is the regulation of lipid metabolism, and fermented cereals have shown promise in mitigating steatosis. Both *in vitro* and *in vivo* studies indicate that fermented cereals can reduce lipid accumulation. For instance, Ai et al. [159] demonstrated in insulin-resistant HepG2 cells that rice bran fermented with *Lactobacillus fermentum* MF423 (FLRB) increased glucose consumption and decreased lipid accumulation, a critical factor in preventing the progression to more severe liver conditions. This was further supported by their animal model study, where FLRB treatment resulted in reduced liver cell damage and fewer lipid droplets. Similarly, Zhao et al. [160] found that fermented barley with *Lactiplantibacillus plantarum* dy-1 (FBBE-PS) alleviated lipid deposition in high-fat HepG2 cells by inhibiting unsaturated fatty acid biosynthesis. These findings are supported by *in vivo* studies such as Guan et al. [161] that showed a decrease in TG, TC, and NEFA in the livers of rats fed a high-fat diet when treated with black barley fermentation broth. These studies collectively suggest that fermented cereals can effectively modulate lipid metabolism, offering a potential strategy for preventing and managing non-alcoholic fatty liver disease (NAFLD).

### 4.2. Effects on Inflammation and Oxidative Stress

Chronic inflammation and oxidative stress are key drivers of liver damage. Fermented cereals have demonstrated the ability to modulate these processes. Zhang et al. [162] reported that fermented barley, rich in VA, reduced the secretion of pro-inflammatory cytokines such as TNF-α and IL-6 in HepG2 cells, which is crucial for preventing chronic inflammation-related liver damage. In animal models, Guan et al. [161] observed increased activities of SOD and GPx and decreased TBARS levels in rats treated with black barley fermentation broth, indicating a reduction in oxidative stress. Additionally, Alharbi et al. [20] found that probiotic-enriched fermented oat extracts (FOE and HFOE) significantly reduced liver function biomarkers (ALT, AST, and ALP) in diabetic rats, further highlighting the hepatoprotective effects of fermented cereals against inflammatory and oxidative damage. These results demonstrate that fermented cereals can mitigate inflammation and oxidative stress, thereby protecting the liver from damage.

### 4.3. Effects on Glucose Metabolism

The regulation of glucose metabolism is also important for liver health. Zhang et al. [162] and Ai et al. [159] both showed that fermented cereals increased glucose consumption in HepG2 cells, a crucial factor in the prevention of MASLD. These *in vitro* studies show a potential mechanism that could be translated into *in vivo* benefits.

## 5. Conclusions

This review highlights the significant benefits of the fermentation process, particularly SSF, in enhancing the health properties of cereals by increasing their bioactive compound content. In addition to the fermentation method, the specific microbes used play a crucial role, as different strains result in varying phenolic enhancements. The increase in these bioactive compounds through SSF contributes to the superior health benefits of fermented cereals compared to their unfermented counterparts. Notably, recent studies show that many of the phenolic compounds enhanced during fermentation possess hepatoprotective effects, as demonstrated in both *in vivo* animal models and *in vitro* studies. Moreover, the review emphasizes the promising potential of fermented cereals in supporting liver health. However, most existing research has been conducted in animals or *in vitro* using HepG2 cells, with a lack of human studies. Overall, ongoing research into the fermentation of cereals and their bioactive compounds holds promise for developing dietary strategies to improve liver health and reduce the global burden of liver diseases.

Therefore, future research should focus on clinical trials in humans to further explore the therapeutic potential of fermented cereals in liver disease prevention and management.

## Figures and Tables

**Figure 1 nutrients-17-00900-f001:**
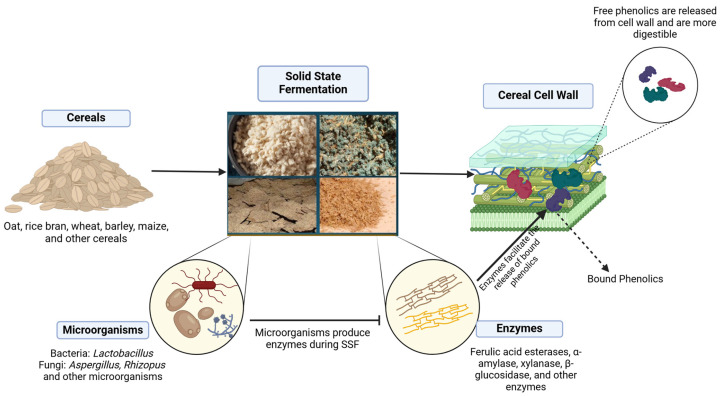
Mechanisms by which solid-state fermentation (SSF) enhances cereal bioavailability through increased free phenolics, which are more readily digestible in the human body.

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
