# Peer review of "Solid-State Fermented Cereals: Increased Phenolics and Their Role in Attenuating Liver Diseases"

_nutrients, 2025, doi:10.3390/nu17050900_

Round 1

Reviewer 1 Report

Comments and Suggestions for Authors

the current manuscript is a comprehensive review article that describes the potential health benefits of fermented cereals, particularly, their phenolic compounds, in managing and preventing liver or ling liver diseases.

The review was conducted well and in good English. My comments for the authors are the following:

The authors should decrease the similarity rate below 20%, it is now 27%. In addition, the authors should provide a graphical abstract with the concept of the study and mostly they could present some data in a figure.

Some other scientific issues that must be given more clearly are a) How do phenolic compounds in fermented cereals impact liver diseases?

b) Are there any gaps in the current literature/research on fermented cereals and liver diseases or liver health?

c) How does fermentation enhance the bioavailability of nutrients in cereals? A possible mechanism in the form of a graphical figure would increase the quality of this review.

Based on the overall quality of this review article, I suggest a minor revision before further consideration for publication.

Author Response

Responses to Reviewer 1

Thank you very much for your valuable comments and they are highly appreciated.
The authors are really grateful for the very pertinent observations made by the referees and editor. The responses to the individual/specific sections or observations in the manuscript have been outlined in the sections ensuing:

a. How do phenolic compounds in fermented cereals impact liver diseases?

Thank you for the question. The specific mechanisms of each phenolic compound in fermented cereals to attenuate liver diseases are discussed in Section 3.2. However, to explained more detailed, the phenolic compound content in cereals is enhanced by the fermentation process, especially solid-state fermentation (SSF). SSF significantly enhances the concentration of phenolic compounds in cereals, including trans-ferulic acid (tFA), p-coumaric acid (pCA), vanillic acid (VA), caffeic acid (CA), syringic acid (SYRA), and gallic acid (GA), among others (as detailed in Section 2.2 and summarized in Table 1). Numerous studies have reported the hepatoprotective effects of these individual phenolic compounds, and these effects are comprehensively reviewed in Section 3.2 and presented in Table 2.

Furthermore, section 4 explores the effects of fermented cereals themselves on liver health and disease. However, the literature directly addressing the impact of whole fermented cereal products on liver health is currently limited, as many studies focus on the effects of isolated phenolics. We summarize the available evidence in this section. This scarcity of research focusing on the complex interplay of the fermented cereal matrix and liver health, particularly in human trials, highlights a critical gap in the current literature. We emphasize this need for future research to specifically address the impact of fermented cereals, rather than isolated phenolics, on liver health.

b. Are there any gaps in the current literature/research on fermented cereals and liver diseases or liver health?

This review identifies several key research gaps:

  1. First, as noted in the introduction, despite reports of the beneficial effects of fermented cereals on various health conditions, there is a notable absence of reviews specifically addressing their benefits and mechanisms in attenuating liver diseases. This review paper addresses this critical gap.
  2. Second, while Section 2.2 identifies Lactobacillus, Aspergillus, and Rhizopus as commonly used microorganisms in cereal fermentation, research directly linking Rhizopus, particularly in cereal products, to liver health is limited, despite its use in SSF of several cereals (Table 1). Future research should investigate the effects of SSF with Rhizopus on cereal products and liver health.
  3. Third, considering the significant impact of SSF, Section 2.2 also highlights the challenges in scaling up this process for industrial applications. These challenges include maintaining uniform growth conditions, controlling mass and heat transfer, and ensuring culture homogeneity. Future research should explore advanced biotechnological tools to engineer more efficient microbial strains. Combining SSF with other processing techniques, such as extrusion or high-pressure processing, to enhance phenolic compound bioavailability is also a promising avenue. Furthermore, integrating genetic engineering to produce microorganisms with enhanced enzymatic capabilities warrants investigation.
  4. Fourth, although the mechanisms by which individual phenolic compounds in fermented cereals attenuate various liver diseases are discussed in Section 3.2, the studies are primarily in vivo (using animal models, mostly rats) and in vitro (using Hep G2 cells), with a lack of human studies. Thus, future studies should focus on human clinical trials. This gap is also noted in the review's conclusion.
  5. Fifth, as mentioned in Section 3.11, while several compounds have been recently explored for their hepatoprotective effects, others (γ-O, EP, RU, QU, and VX) have limited available data regarding their hepatoprotective properties (Table 2). Further research is needed to thoroughly investigate their potential mechanisms and efficacy in liver protection.
  6. Sixth, as discussed in Section 4, research on the effects of fermented cereals on liver health remains relatively limited, with most studies concentrating on the impact of individual phenolic compounds enhanced through fermentation. Thus, future studies need to focus more on the effects of fermented cereals themselves on liver health, especially in human studies. (It's important to differentiate between studying the individual components vs. the whole food matrix)

c. How does fermentation enhance the bioavailability of nutrients in cereals? A possible mechanism in the form of a graphical figure would increase the quality of this review.

Thank you for your suggestion. A graphical figure has been added to the review

Reviewer 2 Report

Comments and Suggestions for Authors

This review highlights the significant benefits of the fermentation process, particularly SSF, in enhancing the health properties of cereals by increasing their bioactive compounds content. It also demonstrates hepatoprotective effects of the phenolic compounds enhanced during fermentation possess in both in vivo animal models and in vitro studies. There are some suggestions for the study:

  1. The abstract section of this review paper could be more concise.
  2. Since the title is “Fermented cereals: increased phenolics and their role in attenuating liver diseases” but this paper just mentions about the liquid state fermentation (LSF) at one place. It’s necessary to compare the SSF and LSF in details to support the SSF’s advantages, or modify the title about the Fermented cereals.
  3. In section 2, I suggest to subdivided into 2.1 the influence of fermentation types, and 2.2 the influence of microorganisms used in cereal fermentation. And other factors can be discussed.
  4. The title of Section 3 is repeated with Section 2.
  5. In table 2, the format could be improved to make the table smaller. In addition, the alteration could be described about the main biomarker.
  6. In section 3.2, the potential influence of cereal-derived polyphenols on enzymes during digestion, gut microbiota and bile acids metabolism could be discussed. Polyphenol may inhibit Bile Salt Hydrolase activity in bile acid transformation and influence liver function. A literature is recommended to be cited and support the effect of bile acids metabolism in liver health: : Cai H, Zhang J, Liu C, et al. High-Fat Diet-Induced Decreased Circulating Bile Acids Contribute to Obesity Associated with Gut Microbiota in Mice. Foods, 2024,13(5).
  7. In section 4.1 and 4.2, there are too much divisions of paragraph, and these cases should be well organized by their logic relationships with transition to support important conclusion, instead of describing them one by one in separate paragraphs.
  8. In line 301, the following content should be introduced in the section 2: the enhancement of the phenolic compounds was observed by different instrument, but mostly with HPLC, HPLC-DAD, UHPLC, or UPLC-Q-TOF-MS, which are known as the fast detector of components.46-48,50,72,101

Author Response

Responses to Reviewer 2

Thank you very much for your valuable comments and they are highly appreciated.
The authors are really grateful for the very pertinent observations made by the referees and editor. The responses to the individual/specific sections or observations in the manuscript have been outlined in the sections ensuing:

  1. The abstract section of this review paper could be more concise.

Thank you for your suggestions. The abstract has been made more concise as requested.

  1. Since the title is “Fermented cereals: increased phenolics and their role in attenuating liver diseases” but this paper just mentions about the liquid state fermentation (LSF) at one place. It’s necessary to compare the SSF and LSF in details to support the SSF’s advantages, or modify the title about the Fermented cereals.

The title has been changed to " Solid-state fermented cereals: increased phenolics and their role in attenuating liver diseases”.

  1. In section 2, I suggest to subdivided into 2.1 the influence of fermentation types, and 2.2 the influence of microorganisms used in cereal fermentation. And other factors can be discussed.

Thank you for your suggestions. The improvements have been implemented.

  1. The title of Section 3 is repeated with Section 2.

The title has been changed to "The enhanced antioxidant activity and phenolic content of SSF cereals for liver health”.

  1. In table 2, the format could be improved to make the table smaller. In addition, the alteration could be described about the main biomarker.

Thank you for your suggestions. We have reduced the table's size by revising the "Pathway/Mechanism of Action" column for brevity. Furthermore, we have added details regarding biomarker alterations.

  1. In section 3.2, the potential influence of cereal-derived polyphenols on enzymes during digestion, gut microbiota and bile acids metabolism could be discussed. Polyphenol may inhibit Bile Salt Hydrolase activity in bile acid transformation and influence liver function. A literature is recommended to be cited and support the effect of bile acids metabolism in liver health: : Cai H, Zhang J, Liu C, et al. High-Fat Diet-Induced Decreased Circulating Bile Acids Contribute to Obesity Associated with Gut Microbiota in Mice. Foods, 2024,13(5).
  2. In section 4.1 and 4.2, there are too much divisions of paragraph, and these cases should be well organized by their logic relationships with transition to support important conclusion, instead of describing them one by one in separate paragraphs.

Thank you for your suggestions. To improve the clarity and flow of Section 4, we have reorganized the content based on logical relationships. Instead of individually discussing past literature on the effects of fermented cereals, we have grouped them into three categories: 1. Lipid metabolism and steatosis, 2. Inflammation and oxidative stress, and 3. Glucose metabolism.

  1. In line 301, the following content should be introduced in the section 2: the enhancement of the phenolic compounds was observed by different instrument, but mostly with HPLC, HPLC-DAD, UHPLC, or UPLC-Q-TOF-MS, which are known as the fast detector of components.46-48,50,72,101

Thank you for your insightful observation. We agree with your suggestion and have moved the sentence to Section 2 to improve the flow and context.